# Regeneration of Horizontal Bone Defect in Edentulous Maxilla Using the Allogenic Bone-Plate Shell Technique and a Composite Bone Graft—A Case Report

**DOI:** 10.3390/medicina59030494

**Published:** 2023-03-02

**Authors:** Zoran Kovac, Tomislav Cabov, Marko Blaskovic, Luka Morelato

**Affiliations:** 1Department of Prosthodontics, Faculty of Dental Medicine, University of Rijeka, 51000 Rijeka, Croatia; 2Department of Oral Surgery, Faculty of Dental Medicine, University of Rijeka, 51000 Rijeka, Croatia

**Keywords:** allogeneic bone graft, biomaterials, shell technique

## Abstract

An insufficient volume of the alveolar bone may prevent implants from being placed in the prosthetically optimal position. Complex restoration of bony structures is required to achieve long-term peri-implant bone stability and represents an adequate prosthetic solution. *Background and Objectives*: The shell technique has become a widespread and important method for guided bone regeneration in dentistry. Allogeneic bone materials appear to be the most similar substitution for autogenous bone transplants. However, there are few studies using cortical bone allografts in combination with a mix of autogenous and xenograft materials for the augmentation of horizontal ridge defects. This combination offers the advantage of reduced patient morbidity while adding adequate volume and contour to the alveolar ridge. *Case report*: The present case study aimed to clinically and radiographically evaluate the efficacy of allogenic cortical bone lamina combined with a composite bone graft in the augmentation of a horizontal bone defect in the edentulous maxilla during a 6-year follow-up period. Three CB CT scans taken before treatment, 6 months after the augmentation period/before implant placement, and after a 6-year follow-up period, were analyzed using stable referent points. After the 6 -year follow-up period, the average resorption rate was 21.65% on the augmented buccal side, with no implant exposure being observed. *Conclusions*: The bone shell technique used in conjunction with allogenic bone plates combined with autogenous bone, xenografts, and collagen membranes is an effective technique to manage horizontal ridge defects.

## 1. Introduction

Tooth loss is associated with the loss of bone and soft tissue structures. The natural remodeling processes of the alveolar socket begin immediately after extraction and may result in up to 50% resorption of alveolar bone in the subsequent 3-month period [1]. This insufficient volume of the alveolar bone may prevent implants from being in the prosthetically optimal position. The restoration of bony structures is sometimes required to achieve long-term peri-implant bone stability and represents an adequate prosthetic solution [2].

For this reason, numerous surgical techniques, such as particle grafts, block grafts, the shell technique, ridge splitting, and distraction osteogenesis, have been used [3].

The shell technique, also called the Khoury technique, uses a cortical bone block harvested from the intraoral donor area. The grafted block is then split into thin cortical plates. Next, these cortical plates are attached to the residual alveolar ridge with osteosynthesis screws to create a contained defect that is then filled with autologous bone chips that have been harvested from the donor site or collected from milling other parts of the obtained block graft in a bone mill [4,5].

In the formed container, the avascular cortical bone plate reduces bone resorption by less than 10%. This results in volume stability, allowing the contour of the alveolar ridge to be restored, achieving predictable results. At the same time, the particulated graft in the container induces accelerated vascularization [6]. Tunkel et al. concluded that because of low resorption rates, simultaneous implant insertion is possible, even in the case of vertical bone augmentation [7].

The short- and long-term results after augmentation via the shell technique show low complication rates and excellent volume stability, even ten years after the procedure [8].

Although many instruments and options are available for intraoral bone harvesting, the need for autologous bone harvesting is a significant negative aspect of the method described above, with disadvantages including a second surgical site with the risk of donor site morbidity and increased pain as well as additional surgical time [9,10].

To avoid bone-harvesting procedures, numerous biomaterials have been marketed for alveolar bone regeneration, with bovine and synthetic bone substitutes dominating the market [11].

Although predictable results have been reported for these materials, they are often inadequate for treating complex bone defects and only function in the form of granules, so their primary use is for socket preservation as well as for the regeneration of small defects [12,13].

Autologous bone transplants have remained the gold standard and the only reasonable option for treating complex cases and large vertical bone defects [5]. Other extraoral donor sites are represented by treatment using calvarial and iliac crest grafts [14,15].

Allogeneic bone materials appear to be the most similar substitution for autogenous bone transplants in clinical applications in terms of patient outcomes [16]. Allografts can be used in either the particulate or block graft form and can be used alone or combined with autogenous, xenogeneic, or alloplastic materials. When evaluating and comparing GBR procedures with allogeneic and autologous bone grafts, several authors have reported comparable results and similar survival and success rates for implants placed in augmented areas [17].

However, after a thorough literature search, we have determined that there are few studies on the use of cortical bone allografts combined with a mix of autogenous and xenograft materials for the augmentation of horizontal ridge defects. Neither of the studies found has a long follow-up period. This combination offers the advantage of reduced patient morbidity while adding adequate volume and contour to the alveolar ridge.

The aim of the present case study was to clinically and radiographically evaluate the efficacy of the use of allogenic cortical bone lamina combined with a composite bone graft in the augmentation of a horizontal bone defect in the edentulous maxilla during a 6-year follow-up period.

## 2. Case Report

A 52-year-old female patient was referred to the Department of Oral Surgery, Dental Clinic of Clinical Hospital Center Rijeka, because of disturbances while wearing an upper total prosthesis. Consistent with the ethical requirements of the Faculty of Dental Medicine, University of Rijeka, Croatia, and the Clinical Hospital Center Rijeka, Croatia, written informed consent for the publication of this study was obtained.

The patient was under periodontologist supervision due to chronic periodontitis for many years before the decision was made to extract the patient’s teeth. The patient did not have any medical comorbidities or allergies. However, she was a smoker, smoking more than ten cigarettes per day. She quit smoking before surgery but returned to the habit one year after. Due to the long edentulous period in posterior regions (more than ten years) and of the high levels of bone resorption caused by periodontitis, there was a lack of bone for correct prosthetic implant placement. Because significant augmentation was needed, it was decided that allogenic bone lamina would be combined with a composite graft. After flap retraction (Figure 1a), augmentations were performed using the shell technique during conscious sedation under anesthesiologist supervision. Three allograft bone laminas, Maxgraft cortico (Botiss gmbh, Berlin, Germany), were used to augment the frontal maxilla (Figure 1b). Laminas were fixed in place with bone fixation screws that were 11 mm long and 1.2 mm in diameter (Ustomed Instrumente, Ulrich Storz GmbH & Co., Tuttingen, Germany). At the left lateral incisor, two 1.6 mm screws were placed to bridge the gap between the laminas and used as a tenting screw. The formed contained defects were filled with a combination of autogenous bone collected with a bone scraper (Safescraper Twist; META, Reggio Emilia, Italy) from the left mandibular ramus and mixed with xenogenic bone (Cerabone; Botiss gmbh, Berlin, Germany) in a ratio of 50%. To augment the lateral part of the maxilla in a molar region, a standard lateral approach sinus lift was performed and filled with the same xenograft material. Three collagen membranes (Jason membrane; Botiss gmbh, Berlin, Germany) were placed to protect the augmented site against epithelial invasion (Figure 1c). Mucosa was sutured with monofilament 5-0 and 6-0 sutures (Nylon; SERAG-WIESSNER GmbH & Co., Naila, Germany). After an uneventful healing period of 6 months, a control cone beam computed tomography (CB CT) scan (3D Accuitomo 170; J. Morita Mfg. Corp., Kyoto, Japan) was carried out before implant placement. During the healing period after the graft placement and after the implant osteointegration period, the patient was instructed to wear a mobile prosthesis without buccal flanges as little as possible. Reopening (Figure 1d) and implant placement were carried out using a bone-supported implant guide planned using 3Shape Implant Studio (3Shape, Copenhagen, Denmark) and that had been 3D-printed on a Form 2 printer (Formlabs, Somerville, MA, USA). Six of the placed implants (Nobel Parallel; Nobel Biocare AB, Göteborg, Sweden) were 11.5 mm long, four frontal implants were 3.75 mm wide, and two implants in tooth positions 16 and 26 were 4.3 mm wide. After the implant’s osseointegration period of 4 months, the implant sites were reopened, and healing abutments were placed. After mucosal healing was achieved around the healing abutments, multiunit abutments (Nobel Biocare AB, Göteborg, Sweden) were placed. A provisional screw-retained bridge was used until a definitive screw-retained bridge was finished. A definitive hybrid screw-retained bridge was crafted using a metal construction on which single zirconia crowns layered with feldspathic ceramic were cemented (Figure 2a) Gingiva was made from a pink composite material (SR NEXCO Gingiva; Ivoclar Vivadent AG, Schwan, Liechtenstein). Follow-ups visits were made annually, and a control CB CT scan was performed and intraoral clinical situation photos were taken after six years (Figure 2b). Before every previously described stage orthopantomographic image was taken (Figure 3).

Three available CB CT scans, those carried out before treatment, 6 months after the augmentation period/before implant placement, and after a 6-year follow-up period, were analyzed. Using Blue Sky Plan software (Blue Sky Bio, LLC, Libertyville, IL, USA), the CB CT scans were superimposed using the same stable referent points on the patient’s zygomatic and pterygoid bone. This allowed us to obtain images showing precisely how the of preoperative and postoperative sites overlapped. Measurements were carried out using the same software and were taken for the six positions at the same distance from the midline at 5 mm, 10 mm, and 15 mm. Measurement directions were the same due to good visualization provided by the overlapped scans. CB CT scans were carried out using the same device with the same exposure parameters, exported in DICOM format, and measured in Blue Sky Plan software (version number 4.7.55).

The measurement results were obtained in six positions at the same distance from the midline, as shown in Table 1. The sites were divided into the groups R1, R2, and R3 on the right side at 5 mm, 10 mm, and 15 mm from the midline; on the left side, sites were labeled as L1, L2, L3 at distances of 5 mm, 10 mm, and 15 mm from the midline. The results were analyzed in Microsoft Excel software (version number 16.69; Microsoft Corporation, Redmond, SAD) to show the augmented site’s average value and resorption rate after a 6-year follow-up period in linear metrics measurements (millimeters) and percentages.

The obtained results show that before the augmentation procedures, the average residual ridge was 5.48 mm. Six months after the augmentation procedures, the measured sites were, on average, augmented by 5.94 mm, indicating average augmentation of 47.9 percent per site. The resorption rate was 21.4% after a 6 years-follow up period. The average width of resorption of the measured sites was 2.45 mm.

## 3. Discussion

Although the shell technique is a bone augmentation technique that is technically sophisticated and demands superior surgical skills, different authors suggest that the significant advantage of the shell technique over autogenous full block transplants is the reduced graft resorption of 5–9% compared to resorption rates of 21–25% for autogenous full-block transplants [18,19].

Several factors may influence resorption rates after bone grafts, such as the type of reconstruction, surgical technique, type of biomaterial, healing time, and, most importantly, the measurement method [20,21].

The measurement methods used in our study are linear, similar to most other studies, and measurements were conducted on superimposed models, so the measurement of every chosen site should be in exactly the same position in all measured models.

In most studies, measurements were taken between 4 and 6 months after augmentation procedures, although clinicians know the bone remodeling process continues throughout one’s entire life.

Our study showed a 6-year follow-up period and a resorption rate of 21.4%, which is more similar to that of autogenous block transplants.

A solution to obtain significantly lower resorption could be additional coverage of the recipient site with bone substitutes with low turnover rates, such as xenogenic bone grafts and resorbable collagen membranes [22]. For example, Maiorana et al. found that using deproteinized bovine bone to cover onlay block grafts reduced resorption by almost 50% compared to in the absence of coverage [23]. A similar result was obtained in the study by Cordaro et al. However, simultaneously, the authors concluded that using bone substitutes and barrier membranes in combination with block grafts increased the frequency of complications [24].

Despite the fact that using a collagen membrane to cover the augmented area could increase the complication rate in the form of dehiscence, the coverage of allogenic grafts is recommended [22].

The allograft shell technique has an additional advantage in the much larger bone volume that can be obtained and used for augmentation with the same harvesting volume due to the partial particularization of the graft. [25].

In the presented case, even more volume is archived by combining the xenograft with an autogenous bone graft. Because of the added xenograft material, we waited 6 months to enhance the vascularization and integration of graft. We did not observe any complications related to block disintegration during the removal of the fixation screws or during implant placement.

All blocks were used in deficient maxillae; blood supply to the maxilla is better than that to the mandible, which may be another reason for the excellent integration during the healing period but could have also been responsible for the greater resorption rate [26].

The primary concern of clinicians is the potential complications that may occur during bone harvesting, especially damage to the inferior alveolar nerve. However, a major prospective clinical trial study with a 10-year follow-up demonstrated that this risk is marginal [9]. In our case, the healing periods were uneventful, and the donor site healed without any complications. Moreover, because the grafted material was only scraped from the ramus using bone scrapers, this method is associated with a low complication rate due to shallow bone harvesting.

The data available on allogeneic cortical struts for the shell technique are very scarce, with only a few reports, and to the best of our knowledge, there are no long-term follow-up studies [27].

Würdinger et al. obtained excellent results and adequate bone quantity to install dental implants. Even in the case of the resorption of the thin allogeneic bone plate after implantation, the implants were not exposed [28]. Tunkel et al. concluded that combining allogeneic bone plates with autogenous bone chips represents a promising alternative to autogenous transplants. Furthermore, they also state that using bone chips has many benefits, as the risk of complications associated with bone harvesting is avoided because bone chips must be collected with a bone scraper to eliminate the risk of nerve lesions [25].

In the presented case study, radiological measurements of six positions 6 years after implant installation in the augmented site using the bone shell technique were retrospectively evaluated and showed long-term stable results comparable to those of the gold-standard method of autogenous block grafting.

## 4. Conclusions

Within the limitations of the present study, it can be concluded that a bone lamina technique using allogenic bone plates in combination with autogenous bone, xenograft materials, and collagen membrane can be used as an effective technique to manage cases of horizontal ridge defects. The key to successfully managing such defects is proper case selection. Further long-term studies with a larger sample size and long-term follow-up are necessary to substantiate the obtained results.

## Figures and Tables

**Figure 1 medicina-59-00494-f001:**
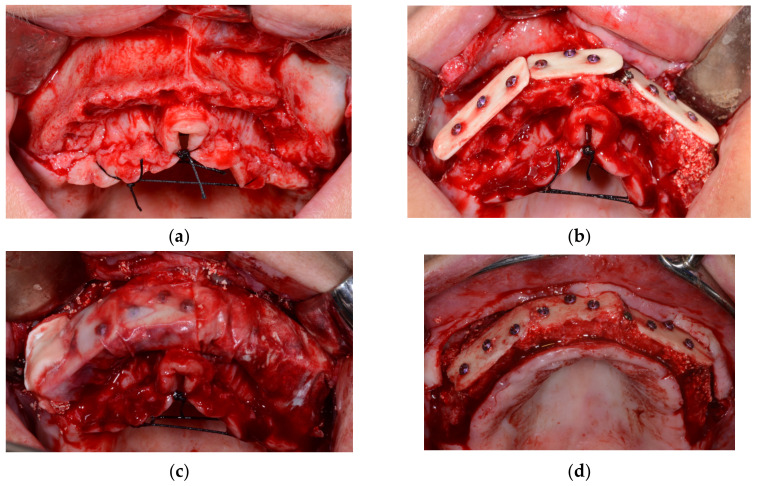
(**a**)—Initial situation after flap retraction; (**b**)—allogenic cortical lamina fixed in place; (**c**)—collagen membranes placed to protect the augmented site against epithelial invasion; (**d**)—augmented ridge after reopening (6 months after augmentation).

**Figure 2 medicina-59-00494-f002:**
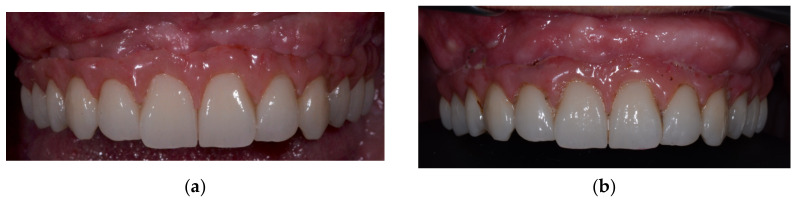
(**a**)—Final prosthetic work in place; (**b**)—final prosthetic work after 6-year follow up period.

**Figure 3 medicina-59-00494-f003:**
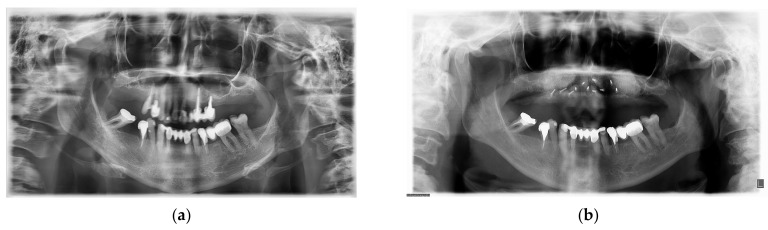
(**a**)—Initial orthopantomographic image taken before teeth extraction; (**b**)—orthopantomographic image after augmentation; (**c**)—orthopantomographic image after implant insertion; (**d**)—orthopantomographic image with implant fixture and definitive prosthetic work in place to check whether there is an adequate passive fit.

**Table 1 medicina-59-00494-t001:** Measurements taken in six positions at the same distance from the midline and calculated changes in the alveolar ridge.

Measurements in mm	R1	R2	R3	L1	L2	L3	Avg	Avg Change (%)
Initial situation	7.06	6.3	2.72	5.85	4.2	6.74	5.48	
Post augmentation (6 months)	12.04	11.71	10.64	12.11	10.72	11.27	11.42	47.9
Follow up (6 years)	9.47	9.97	6.03	9.72	9.47	9.13	8.97	−21.4

## Data Availability

The data presented in this study are available on request from the corresponding author.

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
