# Peer review of "Regeneration of Horizontal Bone Defect in Edentulous Maxilla Using the Allogenic Bone-Plate Shell Technique and a Composite Bone Graft—A Case Report"

_medicina, 2023, doi:10.3390/medicina59030494_

Round 1

Reviewer 1 Report

General Comments

The authors are congratulated for reporting on their clinical work. This manuscript is meant to report on the Restoration of horizontal alveolar bone loss using the shell technique. In its present form, considerable improvements in the write-up are necessary.

Section-wise comments

The Title:      

i)                 The length is Ok

ii)               The title is informative.

iii)             The title does not say it’s a case report, thus if someone reads the article in a search engine, it may be taken as it’s a crossectional study.

Recommendation: Please rephrase the title to include the word case report.

Abstract:       

       i.           Gives summary of what is in the manuscript.

     ii.           The aim of this case report is clear.

   iii.           Grammatical errors were noted.

    iv.           The conclusion is acceptable

Recommendation: Please improve the abstract and English.

Keywords: 

       i.                     The keywords are provided and are appropriate.

Recommendation: None.

Introduction:

       i.                     The introduction does inform the reader about the case, as it is focused, The aim for reporting this case is stated.

     ii.                     The rationale for this case report is stated.

   iii.                     The flow of information within and between the paragraphs is acceptable except in a few paragraphs where the link between 2 paragraphs is missing. E.g paragraphs 2 and 3, and between 3-4.

    iv.                     Grammatical errors were noted.

      v.                     What does GRB stand for?

Recommendation: The authors have very relevant information, however, the information is not smoothly flowing between some paragraphs. The authors can re-arrange the paragraphs for a smooth flow of information by linking one paragraph to another. Please improve the grammar as well.

Case Report   :          

         i.           The authors decided to use the style of a research article to write up a case report. This is rather bizarre.

To be fair and be considerate, the comments on the section of material and method, and results  will be considered as case presentation:

       ii.           The information give is significant.

     iii.           There are significant grammatical errors through out the case presentation section: e.g. mixing of tenses, wrong article etc.

     iv.           When did the patient first report to the authors? It is worth reporting.

       v.           It is worth reporting for how long the patient had been edentulous and if she had any co-morbid conditions, was she a smoker or not?

     vi.           The radiological images of the patient could also be included e.g an Orthopantomogram before and after the procedure. The Overlapping CT image is not enough.

Recommendation: The authors are advised to re-read the authors' instructions on how to report a case report. There is a need to improve the English used. The case presentation requires to be re-structured and additional information is needed.

Discussion:    

       i.           The flow of information is confusing especially in the first 3 paragraphs.

     ii.           In the 4th paragraph, the authors are talking about the vascularization of the graft, and they are discussing graft resorption when it comes to their case, this is confusing.

   iii.           Grammatical errors noted.

    iv.           The authors have tried to discuss their case.

Recommendation: Though the authors have good and valid points for discussion, they are failing to write and discuss their case clearly in most of the discussion section. The English need to be improved.  The discussion requires restructuring.

Conclusion:

       i.           The conclusion is acceptable based on the case presented.

     ii.           The second paragraph of the conclusion is irrelevant for it's more like a discussion.

Recommendation: there is room for improvement

References and citations:    

       i.                     The authors have cited most of the points.

     ii.                     A total of 40 citations for a simple case report is too much.

   iii.                      References have been written uniformly.

Recommendation: the authors need to restructure their manuscript as some of the cited points may be redundant.

Structure and length:          

       i.           It is a lengthy paper.

     ii.           The article is not well-organized, though it is balanced.

Recommendation: The article requires being reorganized.

Logic:            

       i.           The article has been written clearly, though there are some unlinking paragraphs

     ii.           The rationale for this report is clear thus it makes it easy to grab the information.

   iii.           There is no logical consistency between some of the paragraphs in both the introduction and discussion

Recommendation: The authors are requested to make some adjustments to their manuscript

Figures:         

         i.           The images are of fair quality.

       ii.           Figures 1-4, and 5-6 can be combined to form 1 collage with a,b,c, and d, since they present the same thing at different duration.

     iii.           Why no clinical image of the Implant inserted? , and a radiograph of the jaw with implants will be great.

     iv.           Figure 7 is very confusing. There is no key, the colours overlapping are almost similar, and worse enough there is no legend to figure for a reader to follow what authors are trying to portray in this particular figure.

Recommendation: if the authors have better image of the patient (figure 2) then they may consider replacing the current one. Some images can be collaged to reduce the number of figures.

Tables:           

         i.           The use of table 1 is obvious, but table 2 is redundant.

Recommendation: Information in table 2 can either be combined in table 1, or just presented in word format.

English:         

       i.           The English used in the article is not standard.

Recommendation: There is room for improving the English used.

Reviewer 2 Report

Many thanks for the paper submission. This is an intersting paper regarding the regeneration of horizontal bone defect in edentulous maxilla using the allogenic bone plates shell technique and composite bone graft. However some modifications are required.

1) at line 47 the authors should cite others donor sites, please add this phrase

".. Others extraoral donor site are represented by calvarial graft and iliac crest graft.."

please cite the following

Sassano P, Gennaro P, Chisci G, Gabriele G, Aboh IV, Mitro V, di Curzio P. Calvarial onlay graft and submental incision in treatment of atrophic edentulous mandibles: an approach to reduce postoperative complications. J Craniofac Surg. 2014;25(2):693-7. doi: 10.1097/SCS.0000000000000611. PMID: 24621726.   Valentini V, Gennaro P, Aboh IV, Longo G, Mitro V, Ialongo C. Iliac crest flap: donor site morbidity. J Craniofac Surg. 2009 Jul;20(4):1052-5. doi: 10.1097/scs.0b013e3181abb21d. PMID: 19634214.   2) at line 65 the authos should add this phrase:   ".. in fact their major use is for socket preservation and regeneration of small defect.."   please cite the following:   Chisci G, Fredianelli L. Therapeutic Efficacy of Bromelain in Alveolar Ridge Preservation. Antibiotics (Basel). 2022 Nov 3;11(11):1542. doi: 10.3390/antibiotics11111542. PMID: 36358197; PMCID: PMC9687015.     3) the chapter 2. material and methods should be renamed as case report   4)images: the case is well documented and the figures are of adequate quality: I suggest to add a figure with implant fixture inserted   5) the case is well described. Did the patient used a provisional between the time of graft and the time of implant loading?

Round 2

Reviewer 1 Report

General Comments

Thank you for making revisions.

Section-wise comments

The Title:      

i)                 Though the title does say it’s a case report now, however, the way it reads is not appealing.

Recommendation: Please remove the “6 year follow up”:   The title can simply read: Restoration of horizontal alveolar bone loss in an edentulous patient using the shell technique- a case report

Abstract:       

       i.           The main issue with the abstract was the grammar and English used. There is some improvement but still, grammatical errors are there.

Recommendation: Please seek help of an English expert.

Introduction:

       i.                     The flow of information within and between the paragraphs has not been improved. What I have noticed is that the authors have tried to rearrange words in the same sentence, without any effort of linking the information.

     ii.                     Grammatical errors persist.

   iii.                     Long form of GRB has been provided

Recommendation: Please re-arrange the paragraphs and not words within the sentence for a smooth flow of information. Please improve the grammar as well.

Case Report   :          

         i.           The authors have done massive improvement, however, grammatical errors are persistent.

Recommendation: I presume that the authors may not be from a native English-speaking region, as such, kindly seek help from an English language expert in your locality to edit your work.

Discussion:    

       i.           The flow of information is improved.

     ii.           Grammatical errors noted persist.

Recommendation: Please refer to the recommendation above regarding the improvement of grammar.

References and citations:    

       i.                     References improved and were reduced to 28.

Recommendation: None

Structure and length:          

       i.           The article is still not well-organized.

Recommendation: The article requires being reorganized.

Logic:            

       i.           There is still no logical consistency between few of the paragraphs in both the introduction and discussion

Recommendation: The authors are requested to make some adjustments to their manuscript. I believe with the help of an English expert, such illogical consistencies will be taken care of.

Figures:         

         i.           Authors have collaged some images, which is good. Still previous images persist in the document.

Recommendation: The final document should be with collaged images only

English:         

       i.           The English used in the article is not standard.

Recommendation: There is room for improving the English used.

Overall Comments: MAJOR English corrections before acceptance.

Reviewer 2 Report

Congratulations, accept

Author Response

Thank you for your kind comment.